# Identification of Candidate Genes for Endometriosis in a Three-Generation Family with Multiple Affected Members Using Whole-Exome Sequencing

**DOI:** 10.3390/biomedicines13081922

**Published:** 2025-08-06

**Authors:** Carla Lintas, Alessia Azzarà, Vincenzo Panasiti, Fiorella Gurrieri

**Affiliations:** 1Research Unit of Medical Genetics, Department of Medicine, University Campus-Biomedico of Rome, 00128 Rome, Italy; v.panasiti@unicampus.it (V.P.); gurrieri@unicampus.it (F.G.); 2Operative Research Unit of Medical Genetics, Fondazione Policlinico Universitario Campus Bio-Medico, 00128 Rome, Italy; a.azzara@unicampus.it; 3Operative Research Unit of Dermatology, Fondazione Policlinico Universitario Campus Bio-Medico, 00128 Rome, Italy

**Keywords:** familial endometriosis, whole-exome sequencing, germinal, causative genes

## Abstract

**Background**: Endometriosis is a chronic inflammatory condition affecting 10–15% of women of reproductive age. Genome-wide association studies (GWASs) have accounted for only a fraction of its high heritability, indicating the need for alternative approaches to identify rare genetic variants contributing to its etiology. To this end, we performed whole-exome sequencing (WES) in a multi-affected family. **Methods**: A multigenerational family was studied, comprising three sisters, their mother, grandmother, and a daughter, all diagnosed with endometriosis. WES was conducted on the three sisters and their mother. We used the enGenome-Evai and Varelect software to perform our analysis, which mainly focused on rare, missense, frameshift, and stop variants. **Results**: Bioinformatic analysis identified 36 co-segregating rare variants. Six missense variants in genes associated with cancer growth were prioritized. The top candidates were c.3319G>A (p.Gly1107Arg) in the *LAMB4* gene and c.1414G>A (p.Gly472Arg) in the *EGFL6* gene. Variants in *NAV3, ADAMTS18, SLIT1*, and *MLH1* may also contribute to disease onset through a synergistic and additive model. **Conclusions**: We identified novel candidate genes for endometriosis in a multigenerational affected family, supporting a polygenic model of the disease. Our study is an exploratory family-based WES study, and replication and functional studies are warranted to confirm these preliminary findings.

## 1. Introduction

Endometriosis (EMS) is a chronic inflammatory condition characterized by the presence of endometrial-like tissue outside the uterus, commonly affecting the ovaries, fallopian tubes, outer uterine surface, and other pelvic organs. This ectopic tissue responds to hormonal fluctuations during the menstrual cycle, leading to inflammation, pain, and potential adhesion formation. Clinical manifestations include chronic pelvic pain, dysmenorrhea, dyspareunia, abnormal or heavy menstrual bleeding, pain during bowel movements or urination (particularly during menstruation), chronic inflammation, irregular periods, infertility, fatigue, bloating, and nausea. Symptom severity varies among individuals, with some remaining asymptomatic [1].

The exact etiology of endometriosis remains unclear, but several theories have been proposed, including retrograde menstruation, coelomic metaplasia, and immune system dysfunction. Genetic, hormonal, and environmental factors also appear to contribute to disease susceptibility. The diagnostic process is often delayed, sometimes taking years, due to the nonspecific nature of symptoms and the overlap with other gynecological or gastrointestinal disorders. Diagnosis is typically confirmed via laparoscopy, a minimally invasive surgical procedure, although non-invasive imaging techniques and biomarker research are gaining traction [1].

Endometriosis can significantly impact a patient’s quality of life, both physically and emotionally. Chronic pain and infertility may lead to anxiety, depression, and social withdrawal. Thus, early recognition and a multidisciplinary treatment approach—including hormonal therapy, pain management, surgical intervention, and psychological support—are essential for optimal care and long-term outcomes [1].

Diagnosis typically involves pelvic examination, transvaginal ultrasound, or magnetic resonance imaging (MRI). Laparoscopy is also part of the diagnostic workup, even though it is an invasive procedure [2].

Treatment strategies encompass anti-inflammatory medications (e.g., ibuprofen), hormonal therapies (e.g., oral contraceptives and progestins), and surgical interventions (e.g., laparoscopic excision or hysterectomy in severe cases). These approaches aim to reduce pain, suppress ectopic endometrial tissue activity, and improve quality of life. Hormonal therapies function by altering the hormonal milieu to prevent the cyclical stimulation of endometrial-like tissue. Options include gonadotropin-releasing hormone (GnRH) agonists and antagonists, which induce a temporary hypoestrogenic state, thereby reducing lesion size and symptom severity. However, such treatments may have side effects, including bone density loss and menopausal symptoms, often necessitating “add-back” therapy [2].

Surgical treatment, especially laparoscopic excision of visible lesions, is considered in cases resistant to medical therapy or when anatomical distortion compromises fertility. Hysterectomy, with or without bilateral salpingo-oophorectomy, may be considered as a last resort for women with intractable pain who do not desire future fertility [2].

Lifestyle modifications, including stress reduction, anti-inflammatory diets rich in omega-3 fatty acids and antioxidants, and regular physical activity, may also alleviate disease severity and support overall well-being. Complementary therapies such as acupuncture, yoga, and pelvic floor physiotherapy have shown promise in symptom management. Ultimately, an individualized, multidisciplinary approach that considers patient goals, fertility desires, and comorbid conditions is essential for optimal long-term management. [2].

Despite being considered benign, endometriosis can lead to infertility, significant impairment of the quality of life due to chronic pelvic pain, and, in some cases, can progress to cancer. Epidemiological data indicate a higher prevalence of endometrial cancer among women with endometriosis compared to the general population, suggesting endometriosis as a potential risk factor [3]. Both conditions share characteristics such as cellular invasion, neovascularization, unregulated growth, estrogen dependence, apoptosis resistance, chronic inflammation, and metastatic potential [3,4]. Increased cytokine levels, including IL-1β, IL-6, IL-28, and tumor necrosis factors like CCL2, CCL5, and VEGF, have been implicated in the progression of endometriosis to malignancy. An increased presence of macrophages and lymphocytes in peritoneal fluid during endometriosis progression has been observed, along with a documented imbalance between Th1 and Th2 cells [5]. These dysregulated processes are influenced by hormonal imbalances, notably elevated estrogen production and progesterone resistance.

Globally, approximately 10–15% of women of reproductive age are affected by endometriosis [6]. The condition is multifactorial, arising from the interplay between genetic predisposition and environmental factors [7]. While its exact etiology remains elusive, genetic, epigenetic, hormonal, oxidative stress, and immune components are critical contributors. Immune dysregulation, involving inflammatory mediators, cytokines, and immune cells, facilitates the implantation, proliferation, angiogenesis, and development of ectopic endometrial stromal cells [1,2].

Environmental risk factors include prolonged exposure to endogenous estrogens (e.g., early menarche or late menopause), hormonal fluctuations, short menstrual cycles, heavy menstrual bleeding, dietary influences, physical inactivity, gut and vaginal microbiota composition, and smoking. Notably, smoking appears to have an inverse association with endometriosis risk, though findings are inconsistent across studies [8].

The genetic basis of endometriosis is supported by various studies. Twin studies reveal a higher concordance rate among monozygotic twins compared to dizygotic twins. Familial clustering has been documented since the 1950s, with first-degree relatives of affected women exhibiting a five- to seven-fold increased risk [9,10,11]. Familial cases often present earlier onset and more severe symptoms than sporadic cases [12].

Genome-wide association studies (GWASs) have identified significant associations between single-nucleotide polymorphisms (SNPs) in genes involved in sex steroid pathways, including *ESR1*, *GREB1*, *FSHB*, and *CCDC170* [13]. Family-based linkage studies in Australian and British cohorts have pinpointed linkage regions on chromosomes 10q26, 7p13–15, and 20p13. Additional risk loci on 7p15.2 and 1p36.12 have been identified through GWAS. The subsequent resequencing of candidate genes has largely failed to uncover monogenic causes, with the exception of high-penetrance variants in *NPSR1* [14]. Overall, these studies have demonstrated that even in familial cases of EMS, multiple genes contribute to disease susceptibility, highlighting the polygenic nature of EMS.

Furthermore, epigenetic mechanisms play a significant role in the onset of EMS. Genes encoding enzymes involved in estrogen metabolism exhibit promoter hypermethylation, leading to reduced estrogen degradation [15,16]. Conversely, genes such as ESR2, which are involved in estrogen synthesis, display hypomethylated promoters, resulting in elevated estrogen levels. Additionally, the characteristic decrease in progesterone levels observed in EMS is associated with aberrant DNA methylation and histone modifications of the progesterone receptor gene (PGR). Recent studies have also indicated that specific microRNAs and long non-coding RNAs may be dysregulated, further contributing to increased estrogen levels in EMS [17].

Given the complex etiology of EMS and the estimated heritability at 50% based on genome-wide association studies (GWASs) and linkage analyses, we conducted whole-exome sequencing (WES) in a three-generation family affected by EMS to identify potential candidate genes associated with the disease. This approach is particularly powerful in detecting rare variants for complex disorders like endometriosis because it allows the identification of inherited, disease-associated mutations by comparing affected and unaffected family members, reducing background genetic noise. We hypothesize that rare variants co-segregating in affected family members may contribute to EMS.

## 2. Materials and Methods

### 2.1. The Multiplex Family

A multigenerational family was referred for genetic counseling due to a high incidence of endometriosis among its members (Figure 1). Three sisters (T., 51 years old; M., 48 years old; and B., 44 years old), their mother, grandmother, and the daughter of one of the sisters were all affected. Additionally, the mother was diagnosed with breast cancer at the age of 65. On the maternal side, three uncles died of lung cancer, possibly in association with environmental risk factor exposure during their lives. The father, currently 75 years old, has been diagnosed with senile dementia. On the paternal side, an aunt had breast cancer, and her son was diagnosed with both melanoma and sarcoma at the age of 44. The paternal grandfather also had melanoma.

### 2.2. Whole-Exome Sequencing

Whole-exome sequencing (WES) was performed on the three affected sisters and their mother. Genomic DNA was extracted from peripheral blood leukocytes, and WES was commissioned to an external service (DanteLabs SRL, Aquila, Italy) using the Illumina platform with an average coverage of 100×. Details of the lab wet protocol can be found in Modi et al. 2021 [18].

### 2.3. Bioinformatic Analysis

FASTQ files were generated for each individual and processed using the Galaxy online platform [19] to produce individual VCF files for each proband, as well as a combined VCF file containing variants co-segregating among the four affected women. Initially, paired reads were mapped using BWA [20] (human GRCh37/hg19), followed by duplicate removal and variant calling using FreeBayes. version 1.3.7. In our WES analysis, we ensured high data quality with over 90% of the bases exceeding Q30 and coverage uniformity above 80%, supporting reliable and consistent variant detection across the exome. During our WES analysis, each individual showed ~20,000–25,000 raw variants, which were reduced to ~15,000–20,000 after quality filtering (e.g., depth, genotype quality, or call rate), ~5000–10,000 after functional annotation (e.g., selecting exonic, splice-site, nonsynonymous, or LoF variants), ~100–1000 rare variants after frequency filtering (e.g., MAF < 0.01 or absent in population databases like gnomAD), 36 variants after applying co-segregation model, and finally 6 candidate variants after biological prioritization. Variant annotation was carried out using the enGenome-Evai software version v.0.7 [21].

Filtering criteria included coding and splicing variants with coverage ≥20 and a minor allele frequency (MAF) ≤ 0.01. We selected this coverage as it ensures sufficient read depth to reliably detect heterozygous variants, minimize false positives and negatives, and improve the accuracy of variant calling, which is critical for identifying rare pathogenic mutations. MAF was set to ≤0.01 as we wanted to concentrate on rare variants. Initially, exome data were cross-referenced with a curated list of 49 EMS-associated genes derived from the literature [22]. Exome data were also cross-matched with GWAS data on endometriosis [13]. As no variants in the target panel were shared among the four probands, we proceeded to analyze the gene variants shared by the four probands to identify novel candidate genes.

Variant prioritization was performed using Varelect (https://varelect.genecards.org/about/ accessed on 20 September 2024) based on a gene list derived from the analysis and incorporating the following key terms: “oxidative stress,” “inflammation,” “growth,” “adhesion,” and “estrogen.” These terms were selected from the literature. All bioinformatic analyses were conducted according to best practice guidelines [23,24]. In silico prediction tools for missense variants were accessed via Varsome, and CADD, REVEL, and PaPI scores were used to assess potential pathogenicity (https://varsome.com/ accessed on 22 September 2024). CADD, REVEL, and PaPI are computational scores used to predict the pathogenicity of genetic variants, particularly single-nucleotide variants and missense mutations. CADD (Combined Annotation Dependent Depletion) integrates multiple annotations to estimate deleteriousness, REVEL (Rare Exome Variant Ensemble Learner) combines outputs from several tools to assess missense variant pathogenicity, and PaPI (Pathogenicity Prediction Index) uses machine learning on variant and sequence features to classify potential pathogenic effects. In variant prioritization, thresholds for the indices were set to CADD > 20, REVEL > 0.5–0.7, PaPI > 0.5. For the in silico tools: SIFT < 0.05 (deleterious), PolyPhen-2 > 0.85 (probably damaging), and MutationTaster predictions labeled as “disease-causing” with high confidence. Variant frequencies were evaluated using the GnomAD database version 21.1 (https://gnomad.broadinstitute.org/ accessed on 22 September 2024). GnomAD is the database related to variant frequency in the general (healthy) population.

The study was approved by the local ethics committee (Institutional Review Board approval no. 04.21) and conducted in accordance with the Declaration of Helsinki. Written informed consent was obtained from all participants, and data were handled in compliance with ethical regulations.

## 3. Results

The flowchart (Figure 2) below summarizes the process of variant selection.

Following the completion of the bioinformatics pipeline, we excluded variants located in the 3′ and 5′ untranslated regions (UTRs), intronic regions, synonymous variants, and those with low quality or coverage (≤20×). Rare copy number variants were absent.

The 14 genome-wide significant SNPs reported by Sapkota et al. [13] were not present in our exomes.

We then compared the exomes of the four affected individuals against a curated list of 49 EMS-associated genes derived from the literature [22]. No shared variants were identified within this gene panel. Subsequently, we identified 36 variants common to all four patients. These variants were prioritized using the Varelect v 5.21 software, employing the following keywords: “oxidative stress,” “inflammation,” “growth,” “adhesion,” and “estrogen.”

Each gene variant was individually evaluated based on the following:Minor allele frequency (MAF) and number of homozygotes, as per the GnomAD v2.1.1 database (https://gnomad.broadinstitute.org/ accessed on 22 September 2024).Predicted functional impact using in silico tools such as SIFT, MutationTaster, and PolyPhen, alongside integrated scores including CADD, REVEL, and PaPI indices.Biological relevance to processes like growth, inflammation, adhesion, and oxidative stress, assessed through a literature review.Tissue-specific expression profiles, utilizing data from the GTEx database (https://www.gtexportal.org/home/aboutAdultGtex V10 accessed on 22 September 2024).

Based on these criteria, we compiled a list of the most promising candidate variants (Table 1). Except for *MLH1*, none of the genes listed in Table 1 is currently associated with known clinical conditions.

The top EMS candidate gene identified is *LAMB4* (OMIM*616380), which encodes laminin beta 4. This protein is highly expressed in the skin and at lower levels in other tissues. Frameshift mutations in *LAMB4* and related laminin genes have been observed in gastric and colorectal cancers [25]. A significant proportion of these cancers with *LAMB4* mutations also exhibited loss of *LAMB4* protein expression. Laminins are crucial for maintaining normal epithelial structures and play roles in tumor invasion and metastasis, functioning as tumor suppressor genes. Germline truncating variants in *LAMB4* have been detected in individuals with colorectal cancer, with somatic second hits confirming the biallelic inactivation characteristic of tumor suppressor genes [26]. Wang et al. [27] identified another laminin gene, *LAMA4*, as an EMS-associated gene through bioinformatics analysis, while Santin et al. [22] reported *LAMA5* as an EMS candidate gene. The missense variant c.3319 G>A (p.Gly1107Arg) in *LAMB4*, detected in all four probands, has been previously reported once in the ClinVar database without association to a specific phenotype and is exceedingly rare in the general population (MAF: 3/251,316). In silico tools and high CADD, REVEL, and PaPI scores suggest this variant may impact protein function (Table 1).

The second most attractive candidate is *EGFL6* (OMIM*300239), encoding epidermal growth factor-like domain 6, expressed in cutaneous and visceral adipocytes, as well as in the vagina, cervix, breast, and lung. EGFL6 has been shown to promote asymmetric division, maintenance, and metastasis of ALDH+ ovarian cancer cells [28]. More recently, Garrett et al. [29] demonstrated that EGFL6 activates the MAPK pathway, enhancing cancer cell proliferation and migration in vitro, and increasing tumor growth in vivo. These experiments utilized endometrial cancer cells and tissues, with EGFL6 knockdown resulting in reduced tumor growth. The c.1414 G>A (p.Gly472Arg) missense variant identified in the three sisters and their mother is rare in the general population (MAF: 5/183,514). In silico prediction tools and pathogenicity indices suggest this variant may function as a gain-of-function mutation, promoting abnormal growth and migration (Table 1). This variant is listed as a variant of uncertain significance (VUS) in the ClinVar database, with only one submission and no specific condition associated.

The third candidate gene, *NAV3* (OMIM*611629), functions as a tumor suppressor in breast cancer [30] and is expressed in the vagina, uterus, cervix, and other tissues. The c.7154 T>C (p.Leu2385Pro) variant is rare (MAF: 139/275,880) and has not been reported in ClinVar. In silico tools and pathogenicity indices suggest this variant could affect protein function, particularly if a second somatic mutation occurs. *NAV3* is implicated in cell migration through microtubule stabilization and inhibition of random migration [31]. Loss of NAV3 expression has been associated with melanoma and poor survival in breast and nervous system tumors.

The *ADAMTS18* (OMIM*607512) and *SLIT1* (OMIM*603742) variants exhibit lower REVEL scores (0.088 and 0.132, respectively) and less compelling in silico predictions. However, both genes are involved in growth regulation and the identified variants are absent from the GnomAD database, suggesting to keep a potential role in EMS onset. Specifically, *ADAMTS18* has been implicated in tumorigenesis in melanoma and colorectal cancers [32], while SLIT1 functions as a tumor suppressor in breast epithelium [33].

**Table 1 biomedicines-13-01922-t001:** List of the best candidate co-segregating variants with their associated indices and characteristics.

Gene	Gene Function	Variant	MAF	ClinVar	REVEL	CADD	PaPI	In Silico Tools: P/VUS/B+LB	CONS Score	Expression	Expression Levels in Uterine/Endometrial Tissues	Main Reference
LAMB4	Cell adhesion, growth, and migration (oncosuppressor)	NM_007356.3: c.3319G>A, Gly1107Arg	3/251,316	VUS (1) RCV004078131	0.715	24.6	1.0	10/9/8	4.512	Skin, but also other tissues	low	[26]
EGFL6	Growth factor (oncogene)	NM_015507.4: c.1414G>A, Gly472Arg	5/183,514	VUS (1) RCV004071274	0.367	23.5	1.0	10/9/8	5.076	Cutaneous and visceral adipocytes, vagina, cervix, breast, and lung	moderate	[29]
NAV3	Regulates microtubule dynamics, inhibits migration, and restrains dissemination of tumors (oncosuppressor)	NM_001024383.2: c.7154T>C, Leu2385Pro	139/275,880	/	0.425	27.8	1.0	9/7/5	7.844	Ovary, vagina, uterus, and cervix, but also other tissues	low	[30]
ADAMTS18	Proliferation and migration (oncogene)	NM_199355.4: c.1474G>T, Gly492Trp	/	/	0.088	24.1	0.92	-/5/25	3.45	Cervix, breast, cerebellum, and other tissues	moderate	[32]
SLIT1	Growth (oncosuppressor)	NM_003061.3 c.967T>G; Ser323Ala	/	/	0.132	23	0.98	2/6/20	5.514	Mainly brain	low	[33]
MLH1	Mismatch repair (oncosuppressor)	NM_000249.4: c.977T>C; Val326Ala	131/282,782	CIP (2VUS/25B-LB) RCV000035356 (most recent)	0.846	24.5	1.0	8/11/1	7.661	All tissues	high	[34]

**CIP = Conflicting Interpretation of Pathogenicity** for ClinVar classification, and the main accession number is reported, and in brackets number of entries. **MAF** = minor allele frequency as reported in GnomAD including allgenomic ancestry groups; **CONS** = phyloP100 conservation score; **In silico tools** = number of algoritms reprting the variant as P = pathogenetic; VUS = Variant of Unknown Significance; B/LB = b/likely benign; **REVEL** (Rare Exome Variant Ensemble Learner) index is a computational score used in genetics to predict the pathogenicity; **CADD** (Combined Annotation-Dependent Depletion) score is a computational tool used in genetics to estimate the deleteriousness of genetic variants across the genome; **PaPI** (Pathogenicity Assessment of Pathogenicity Indicators) score is a computational metric used to evaluate the pathogenic potential of genetic variants, particularly rare ones; **Expression** = gene tissue expression as reported in the GTEx database (https://www.gtexportal.org/home/ accessed 22 September 2024). In variant prioritization, commonly used thresholds include CADD > 20, REVEL > 0.5–0.7, and PaPI > 0.5. For the in silico tools: SIFT < 0.05 (deleterious), PolyPhen-2 > 0.85 (probably damaging), and MutationTaster predictions labeled as “disease-causing” with high confidence.

The last candidate gene, *MLH1* (OMIM*120436), is well known to be associated with Lynch syndrome and plays a role in the DNA mismatch repair system. The c.977 T>C (p.Val326Ala) variant, shared by all four probands, is rare (MAF: 131/282,782). High CADD, REVEL, PaPI, and pyloP100 scores, along with in silico predictions, suggest potential deleterious effects on protein function. However, multiple functional studies have demonstrated that this variant does not impair the repair system or affect splicing [34]. Consequently, the variant has been classified as benign by expert panels, with the most recent evaluation on 22 December 2024. Though the *MLH1* variant is not considered pathogenetic in the context of Lynch syndrome, we cannot rule out a milder impact contributing to EMS onset, or even a second hit in the endometrial cells, due to the role of the gene *MLH1* in endometrial cancer.

## 4. Discussion

Endometriosis (EMS) is a multifactorial disorder arising from the interplay between environmental exposures and both common and rare genetic variants [1,6]. Among environmental factors, prolonged exposure to endogenous estrogens is paramount, followed by influences such as diet, lifestyle, and physical activity [8]. Genome-wide association studies (GWASs) and linkage analyses have identified several risk loci, including *NPSR1* (neuropeptide S receptor 1), *GREB1* (growth regulation by estrogen in breast cancer 1), *CCDC170* (coiled-coil domain-containing 170), and *ESR1* (estrogen receptor 1). Beyond genetic predisposition, emerging evidence implicates epigenetic dysregulation in EMS pathogenesis [35].

Although EMS is typically considered a benign condition, it possesses the potential for malignant transformation into endometrial, ovarian, thyroid, or breast cancers [3]. The estimated overall risk for such progression is approximately 1%, though this figure may be underestimated [36]. Notably, the candidate genes identified in our study are involved in critical cellular processes such as proliferation, adhesion, migration, and DNA repair. This observation aligns with the findings from somatic DNA analyses of endometriotic tissues, which have revealed recurrent mutations in cancer driver genes, including ARID1A, components of the MAPK/Ras pathway, and the PI3K-AKT-mTOR axis [37].

Our most compelling co-segregating candidate gene is *LAMB4,* which encodes laminin beta 4. Previous studies have proposed other laminin genes, such as *LAMA4* and *LAMA5,* as EMS candidates [22,26,27]. Given the tumor-suppressive functions of laminins [25], we hypothesize that the germline variant identified in our family may predispose individuals to EMS, with a subsequent somatic “second hit” potentially leading to transformation in EMS phenotype. This proposed mechanism mirrors the well-established multi-step model of familial colorectal polyposis driven by germline mutations in the tumor suppressor gene APC, where an initial germline mutation leads to benign polyposis, and a subsequent somatic mutation causes colorectal cancer [38].

Despite the insights provided by GWAS, linkage studies, and twin studies, a significant portion of EMS heritability remains unexplained. To address this gap, whole-exome sequencing (WES) has been employed by various research groups, including ours, to elucidate the complex genetic architecture of EMS.

Several WES studies have aimed to identify rare variants in known EMS-associated genes and to discover novel candidates. For instance, Santin et al. [22] conducted WES on 80 sporadic EMS cases, finding that 43% harbored variants in 13 candidate genes (*FCRL3*, *LAMA5*, *SYNE1*, *SYNE2*, *GREB1*, *MAP3K4*, *C3*, *MMP3*, *MMP9*, *TYK2*, *VEGFA*, *VEZT*, and *RHOJ*); 8.8% had variants in eight additional genes (*KAZN*, *IL18*, *WT1*, *CYP19A1*, *IL1A*, *IL2RB*, *LILRB2*, and *ZNF366*); and 24% carried variants in three novel genes (*ABCA13, NEB,* and *CSMD*). Other family-based WES studies have identified new candidate genes such as *FGFR4, NALCN, and NAV2* [39]; *TNFRSF1B*, *GEN1*, and *CRABP1* [40]; and *CIITA* [41]. Notably, variants in the first three genes were further screened in a larger EMS cohort without identifying additional carriers [39]. None of the candidate gene variants proposed in these studies co-segregated with EMS in our family. Interestingly, one of the candidate genes proposed by Nousiainen et al. [39], *NAV2*, shares significant similarity with *NAV3*, which emerged as our third most promising candidate (Table 1). The variants identified in our candidate genes are all missense mutations, a finding consistent with other studies that have predominantly reported missense variants in EMS-associated genes.

Collectively, our study and others reinforce the polygenic nature of EMS. The five co-segregating variants presented in Table 1 may contribute to EMS pathogenesis through additive or synergistic effects. Among these, *MLH1* is noteworthy; although the variant co-segregating in our family has been classified as benign by expert panels due to its lack of impact on DNA repair and splicing [34], its potential contributory role in EMS cannot be entirely dismissed, especially within a polygenic framework. Supporting this, the variant exhibits high CADD, REVEL, and PaPI scores (Table 1).

The elevated computational pathogenicity scores suggest that, while individually insufficient to cause disease, the *MLH1* variant may act as a genetic modifier in combination with other risk alleles. This aligns with the increasingly recognized model in which complex diseases such as EMS arise from the cumulative effects of multiple low-penetrance variants, each subtly impacting cellular pathways involved in inflammation, immune response, angiogenesis, or hormone regulation. In this context, the presence of high-scoring *MLH1* variants may enhance susceptibility when accompanied by other functionally relevant variants in key regulatory genes. Further functional validation, including transcriptomic and proteomic profiling in affected individuals, will be essential to delineate the biological relevance of these co-segregating variants and their role in modulating disease severity and phenotypic heterogeneity.

In summary, our study has identified five novel EMS top candidate genes co-segregating within a family comprising five affected individuals. Future objectives include (i) functional validation of the identified co-segregating variants (Table 1); (ii) conducting WES in additional families with multiple EMS-affected members to expand the list of candidate genes or confirm any of the genes in our list; (iii) conducting case–control association studies. Upon establishing an extended target panel incorporating candidate genes from other studies, we plan to screen a larger cohort to further elucidate the genetic underpinnings of EMS.

We are aware that our study is limited by the small sample size derived from a single pedigree, the absence of unaffected family controls that restricts the strength of segregation analysis, the lack of replication in unrelated endometriosis cohorts, and the absence of transcriptomic or protein-level validation to support the functional impact of the identified variants. Despite these limitations, the study represents a valid and valuable effort to identify rare, potentially pathogenic variants through a family-based WES.

## Figures and Tables

**Figure 1 biomedicines-13-01922-f001:**
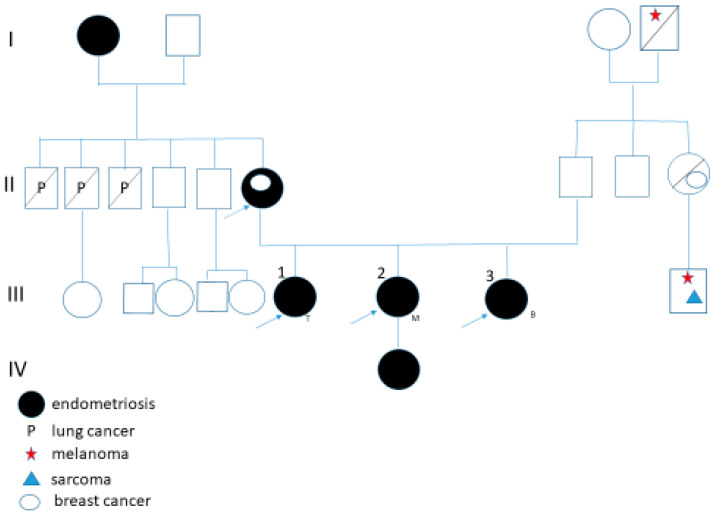
Pedigree of the family. The black symbols represent affected individuals by EMS. WES was performed in the women indicated by the arrow.

**Figure 2 biomedicines-13-01922-f002:**
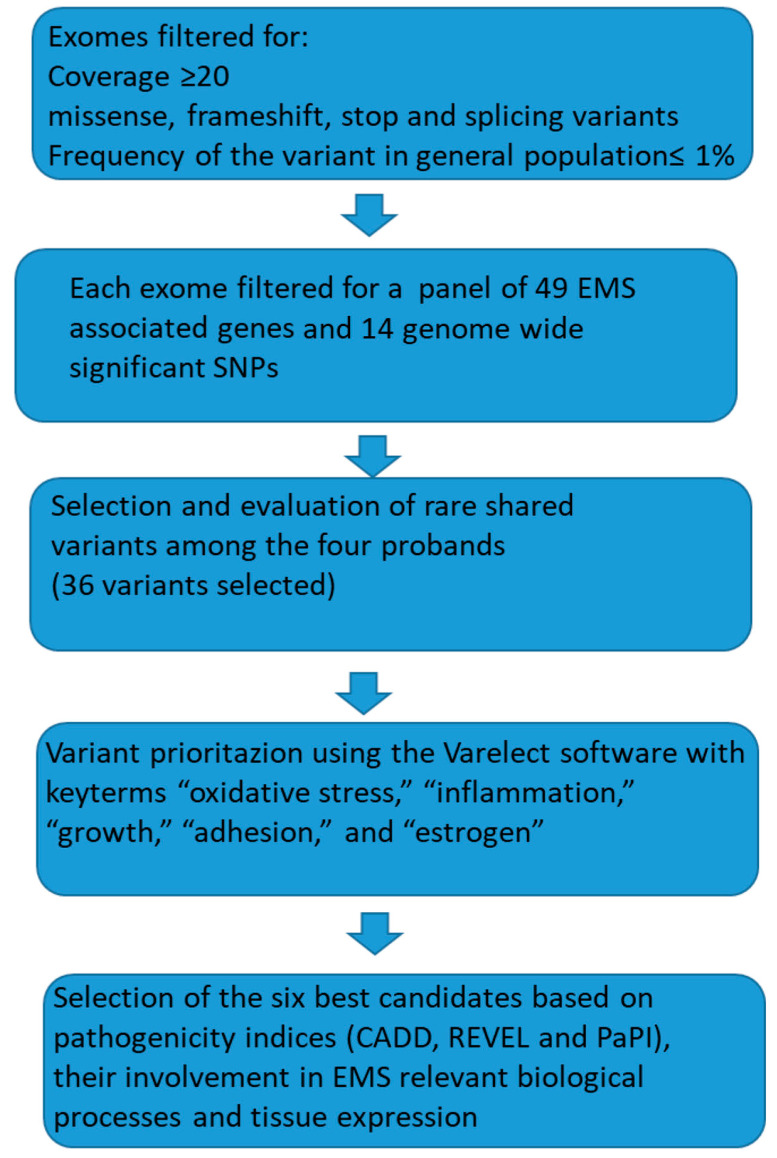
A summary of the workflow performed for the analysis and selection of the six best candidates. The 49 genes for the selected panel are reported in [15] and the 14 genome wide significant SNPs in [13].

## Data Availability

Data are contained within the article.

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
