# Peer review of "Identification of Candidate Genes for Endometriosis in a Three-Generation Family with Multiple Affected Members Using Whole-Exome Sequencing"

_biomedicines, 2025, doi:10.3390/biomedicines13081922_

Round 1
Reviewer 1 Report
Comments and Suggestions for Authors
Comments to the Authors
In the Manuscript titled “Identification of Candidate Genes for Endometriosis in a Three-Generation Family with Multiple Affected Members Using Whole Exome Sequencing” the Authors perform exome sequencing of peripheral blood lymphocytes on four referred patients with a family history of endometriosis (EMS) as well as multiple cancers and identify a set of “36 variants common to all four patients”. Of these, 5 (LAMB4, EGFL6, NAV3, ADAMTS18, and MLH1) are prioritized for further analysis based on specific keywords (“oxidative stress,” “inflammation,” “growth,” “adhesion,” and “estrogen.”), allele frequencies (GnomAD v2.1.1), predicted functional impact (unclear which tools and agreement between different approaches), biological relevance (tools?), and tissue-specific expression (GTEx). None of the 5 candidates were previously identified in EMS by GWAS.
General comments: Previously, a large 11 study GWAS meta-analysis identified 19 SNPs that account for approximately 5% of the variance in EMS suggesting that further exome sequencing studies will likely reveal additional candidates. Thus, the present study extends findings for EMS-relevant alleles in the context of a multi-sibling disease-affected group. Overall, the Manuscript is well-written but needs further improvements with respect to Methods, missing data and informative captions.
Major issues: The Manuscript lack a proper Methods section for the exome sequencing protocol. To simplify matters, it is suggested that a relevant peer-reviewed study that employs similar methods is used as reference while the Authors specify modifications made, if any, to this protocol. Furthermore, Table 1 requires sufficient caption detail so that its content can be appreciated. Lastly, minor illustration suggestions are provided to better capture the findings.
Minor issues: Please address all 23 comments in the attached reviewed Manuscript

Author Response
Dear Reviewer,
Thank you very much for your very useful comments.
We have addressed and included all your suggestions in a new revised copy of the manuscript.
I hope you will appreciate the improvements we made to our manuscript.
Yours Sincerely,
Carla Lintas
In details our responses:
Major issues: The Manuscript lack a proper Methods section for the exome sequencing protocol. To simplify matters, it is suggested that a relevant peer-reviewed study that employs similar methods is used as reference while the Authors specify modifications made, if any, to this protocol. We included the reference of another peer reviewed study that employs similar methods to ours.
Furthermore, Table 1 requires sufficient caption detail so that its content can be appreciated. We included details in the Table 1 caption.
Lastly, minor illustration suggestions are provided to better capture the findings.
Minor issues: Please address all 23 comments in the attached reviewed Manuscript.
We have addressed all the 23 comments in the new version of the manuscript.
Reviewer 2 Report
Comments and Suggestions for Authors
Dear Authors,
I have carefully reviewed your manuscript exploring the genetic underpinnings of familial endometriosis through whole exome sequencing (WES) in a three-generation pedigree. This is a timely and relevant contribution to the field of gynecological genetics and rare disease association studies. To improve the scientific rigor, transparency, and reproducibility of your work, I have evaluated the manuscript according to the STREGA (Strengthening the Reporting of Genetic Association Studies) guidelines. Below, I provide detailed, section-by-section comments and suggestions to help strengthen your manuscript.
Title & Abstract
- Strengths: The title is accurate and informative. The abstract provides a high-level summary of the study and key findings.
- Recommendations:
- Please explicitly mention that this is an exploratory family-based WES study, and that candidate gene findings are preliminary and require replication.
- Include more methodological details in the abstract: number of individuals sequenced, key bioinformatic tools used (e.g., enGenome-Evai, Varsome), and a summary of the variant prioritization pipeline.
Introduction
- Strengths: The introduction provides a comprehensive overview of endometriosis pathophysiology and the known genetic landscape.
- Recommendations:
- Please articulate a clear and testable research hypothesis at the end of the introduction (e.g., that rare coding variants in affected family members may co-segregate and contribute to EMS risk).
- You may briefly highlight why the family-based WES strategy is particularly powerful in rare variant detection for complex disorders like endometriosis.
Methods
This section is central to reproducibility and should fully comply with STREGA standards.
- Participants and Clinical Phenotyping:
- Include detailed clinical phenotypes of affected individuals (e.g., endometriosis stage, age at onset, fertility outcomes).
- Specify diagnostic methods used for EMS confirmation (e.g., laparoscopy, histology), referencing established diagnostic criteria.
- Sample Collection and Sequencing:
- Expand on the DNA preparation, quantification, and library preparation protocol (e.g., capture kit used, sequencing platform and chemistry).
- Include sequence quality metrics (e.g., % bases >Q30, coverage uniformity).
- Bioinformatics and Variant Filtering:
- Provide the exact variant calling pipeline, including tools used for alignment, variant calling, and joint genotyping (e.g., BWA, GATK).
- Justify thresholds for variant filtering (e.g., MAF < 0.01, coverage >20X).
- Please report the total number of variants per individual before and after each filtering step.
- Clarify whether variants on the X chromosome were included and how potential sex bias was addressed.
- Variant Prioritization and Annotation:
- Define thresholds for CADD, REVEL, PaPI, SIFT, PolyPhen, and MutationTaster.
- Explain how keywords used in Varelect were selected (e.g., literature-based vs. ontology-derived).
- Report gene expression data more systematically (e.g., from GTEx), especially in endometrial tissues.
- Ethics and Consent:
- IRB approval is mentioned, but please explicitly state in the main text that informed consent was obtained and that data were handled in compliance with ethical regulations (e.g., GDPR if applicable).
Results
- Strengths: The filtering and prioritization pipeline is logical and well-presented. Candidate variants are appropriately analyzed with in silico predictors.
- Recommendations:
- Report the total number of variants detected per individual, number of co-segregating variants, and final number prioritized.
- Justify why the MLH1 p.Val326Ala variant, which has been classified as benign, is still discussed in the context of EMS.
- Add data on variant zygosity, especially if homozygosity or compound heterozygosity was explored.
- Figure 1 (pedigree): Please expand it to indicate ages at diagnosis, EMS severity (if available), and other comorbidities.
Tables & Figures
- Table 1:
- Define all abbreviations in the table caption.
- Include ClinVar accession numbers and specify the population cohort for MAF data (e.g., European, NFE).
- Add a column indicating GTEx tissue expression scores or rankings in endometrial/uterine tissues.
- Figure Suggestions:
- Consider adding a variant prioritization workflow diagram summarizing your analytic pipeline for visual clarity.
Discussion
- Strengths: The biological interpretation of LAMB4, EGFL6, and NAV3 is insightful and well-supported by literature.
- Recommendations:
- Please moderate language suggesting causality (e.g., “identified gene for EMS”) and instead use more appropriate expressions (e.g., “putative candidate gene associated with EMS”).
- Explicitly acknowledge the lack of functional validation as a limitation.
- Reframe the MLH1 discussion with caution: while REVEL and CADD are suggestive, expert consensus classifies the variant as benign.
- Consider a comparative table of previously published EMS-WES studies to contextualize your findings (e.g., Santin et al., Nousiainen et al., Kina et al.).
Conclusion
- Rephrase strong assertions. Rather than stating that novel genes were “identified,” note that preliminary evidence supports the candidacy of these genes in EMS predisposition.
- Emphasize the need for:
- Functional validation (e.g., gene expression, CRISPR models).
- Segregation analysis in additional family members (if available).
- Screening in larger EMS case-control cohorts.
Limitations
This section is underrepresented and should be more explicitly discussed.
- Please acknowledge the following:
- Small sample size from a single pedigree.
- Absence of unaffected family controls (which limits segregation strength).
- No replication in unrelated EMS cohorts.
- No transcriptomic or protein-level validation.
Language & Terminology
- The manuscript is generally well-written.
- Please revise phrases that imply causation or clinical utility beyond current evidence.
- Ensure consistent terminology (e.g., “variant of uncertain significance” vs. “pathogenic variant”).
References
- Reference formatting is appropriate.
- Please consider citing the STREGA guideline explicitly and reflect alignment with its criteria.
Final Recommendation
This manuscript represents a commendable effort to uncover novel candidate variants in familial endometriosis. However, to meet publication standards for genetic association studies and adhere fully to the STREGA framework, I recommend major revisions. Upon revision, your work may provide a meaningful contribution to the understanding of polygenic and familial EMS etiology.
Thank you for the opportunity to review this work. I look forward to seeing a revised version of the manuscript.
Sincerely,
Author Response
Dear Reviewer,
Thank you very much for your very useful comments.
We have addressed and included all your suggestions in a new revised copy of the manuscript.
I hope you will appreciate our improvements to the manuscript.
Your Sincerely,
Carla Lintas
In details our responses:
Title & Abstract
- Strengths: The title is accurate and informative. The abstract provides a high-level summary of the study and key findings.
- Recommendations:
- Please explicitly mention that this is an exploratory family-based WES study, and that candidate gene findings are preliminary and require replication. We did it.
- Include more methodological details in the abstract: number of individuals sequenced, key bioinformatic tools used (e.g., enGenome-Evai, Varsome), and a summary of the variant prioritization pipeline. We included the number of individuals sequenced and the tools used for analysis.
Introduction
- Strengths: The introduction provides a comprehensive overview of endometriosis pathophysiology and the known genetic landscape.
- Recommendations:
- Please articulate a clear and testable research hypothesis at the end of the introduction (e.g., that rare coding variants in affected family members may co-segregate and contribute to EMS risk). We did it.
- You may briefly highlight why the family-based WES strategy is particularly powerful in rare variant detection for complex disorders like endometriosis. We did it.
Methods
This section is central to reproducibility and should fully comply with STREGA standards.
- Participants and Clinical Phenotyping:
- Include detailed clinical phenotypes of affected individuals (e.g., endometriosis stage, age at onset, fertility outcomes). Unfortunately we do not have this information.
- Specify diagnostic methods used for EMS confirmation (e.g., laparoscopy, histology), referencing established diagnostic criteria. Unfortunately we do not have this information.
- Sample Collection and Sequencing:
- Expand on the DNA preparation, quantification, and library preparation protocol (e.g., capture kit used, sequencing platform and chemistry). We did it.
- Include sequence quality metrics (e.g., % bases >Q30, coverage uniformity). We reported quality metrics as requested.
- Bioinformatics and Variant Filtering:
- Provide the exact variant calling pipeline, including tools used for alignment, variant calling, and joint genotyping (e.g., BWA, GATK). We added a sentence specifying what requested.
- Justify thresholds for variant filtering (e.g., MAF < 0.01, coverage >20X). We justified the chosen thresholds.
- Please report the total number of variants per individual before and after each filtering step. We did it.
- Clarify whether variants on the X chromosome were included and how potential sex bias was addressed. We included variants on X chromosome; sex bias was not considered as our patients were all females.
- Variant Prioritization and Annotation:
- Define thresholds for CADD, REVEL, PaPI, SIFT, PolyPhen, and MutationTaster. Thresholds are now specified in the legend of Table 1.
- Explain how keywords used in Varelect were selected (e.g., literature-based vs. ontology-derived). We specified that keywords were selected from the Literature.
- Report gene expression data more systematically (e.g., from GTEx), especially in endometrial tissues. Expression data were reported from GTEx database. This is now specified in the legend of Table 1.
- Ethics and Consent:
- IRB approval is mentioned, but please explicitly state in the main text that informed consent was obtained and that data were handled in compliance with ethical regulations (e.g., GDPR if applicable). We added a sentence at the end of Materials and Methods section.
Results
- Strengths: The filtering and prioritization pipeline is logical and well-presented. Candidate variants are appropriately analyzed with in silico predictors.
- Recommendations:
- Report the total number of variants detected per individual, number of co-segregating variants, and final number prioritized. We already added this information in Materials and Methods section as requested.
- Justify why the MLH1 p.Val326Ala variant, which has been classified as benign, is still discussed in the context of EMS. We added a sentence in the discussion (about line 288-291) as requested.
- Add data on variant zygosity, especially if homozygosity or compound heterozygosity was explored. We explored all variants as specified in the Materials and Methods section.
- Figure 1 (pedigree): Please expand it to indicate ages at diagnosis, EMS severity (if available), and other comorbidities. We do not have this information.
Tables & Figures
- Table 1:
- Define all abbreviations in the table caption. We did it.
- Include ClinVar accession numbers and specify the population cohort for MAF data (e.g., European, NFE).We did it.
- Add a column indicating GTEx tissue expression scores or rankings in endometrial/uterine tissues. We added a column specifying expression of candidate genes in endometrial/uterine tissues as requested.
- Figure Suggestions:
- Consider adding a variant prioritization workflow diagram summarizing your analytic pipeline for visual clarity. We added a new figure (Figure 2) summarizing our analytic pipeline.
Discussion
- Strengths: The biological interpretation of LAMB4, EGFL6, and NAV3 is insightful and well-supported by literature.
- Recommendations:
- Please moderate language suggesting causality (e.g., “identified gene for EMS”) and instead use more appropriate expressions (e.g., “putative candidate gene associated with EMS”).We did it.
- Explicitly acknowledge the lack of functional validation as a limitation. We did it.
- Reframe the MLH1 discussion with caution: while REVEL and CADD are suggestive, expert consensus classifies the variant as benign. Yes, experts classify it as benign in the context of endometrial cancer not in the context of endometriosis. This is the reason of why we decided to retain the variant. This is now clearly explained in the discussion as requested.
- Consider a comparative table of previously published EMS-WES studies to contextualize your findings (e.g., Santin et al., Nousiainen et al., Kina et al.).
Conclusion
- Rephrase strong assertions. Rather than stating that novel genes were “identified,” note that preliminary evidence supports the candidacy of these genes in EMS predisposition. We did it.
- Emphasize the need for:
- Functional validation (e.g., gene expression, CRISPR models). We did it at the end of the discussion
- Segregation analysis in additional family members (if available). Yes we will consider doing this in the future when other members will become available.
- Screening in larger EMS case-control cohorts. We did it at the end of the discussion
Limitations
This section is underrepresented and should be more explicitly discussed.
- Please acknowledge the following:
- Small sample size from a single pedigree.
- Absence of unaffected family controls (which limits segregation strength).
- No replication in unrelated EMS cohorts.
- No transcriptomic or protein-level validation.
We addressed all these issues concerning limitations of our study in the final paragraph of the discussion.
Language & Terminology
- The manuscript is generally well-written.
- Please revise phrases that imply causation or clinical utility beyond current evidence. We did it
- Ensure consistent terminology (e.g., “variant of uncertain significance” vs. “pathogenic variant”).
References
- Reference formatting is appropriate.
- Please consider citing the STREGA guideline explicitly and reflect alignment with its criteria. STREGA guidelines are essential for observational studies (case-control studies, GWAS studies, cross sectional studies) investigating associations between genetic variants and traits in population-based designs, but are not required for family-based studies using Whole Exome Sequencing (WES), which focus on rare variant discovery through co-segregation rather than population-level association. This is clearly stated in the paper by Little et al., 2009; PMID: 19192285.
- Final Recommendation
This manuscript represents a commendable effort to uncover novel candidate variants in familial endometriosis. However, to meet publication standards for genetic association studies and adhere fully to the STREGA framework, I recommend major revisions. Upon revision, your work may provide a meaningful contribution to the understanding of polygenic and familial EMS etiology.
See previous comment.
Reviewer 3 Report
Comments and Suggestions for Authors
The manuscript titled "Identification of Candidate Genes for Endometriosis in a Three-Generation Family with Multiple Affected Members Using Whole Exome Sequencing" addresses an important topic related to the genetic etiology of endometriosis. The use of whole-exome sequencing in a multigenerational family is a reasonable approach to identify rare pathogenic variants. The authors identified several candidate variants, particularly in LAMB4 and EGFL6. However, while the study presents interesting preliminary findings, the overall methodology and depth of data analysis remain relatively basic. The current version of the manuscript is largely descriptive, and the conclusions are based on limited evidence.
Major Concerns:
The study includes only four individuals for bioinformatic-analysis, which might limit the statistical power and increases the risk of reporting unrelated variants. Although the family-based approach can help mitigate this to some extent, the authors should well discuss the inherent limitations of such a small cohort and avoid misleading the conclusions.
Author Response
Dear Reviewer,
Thank you very much for your useful comments.
In the new version of the manuscript we have addressed all your comments, we have improved the introduction, the results, the methods and the figures and tables as you requested. We have also added a new figure (Figure 2) with the analysis workflow.
I hope you will appreciate our improvements.
Yours Sincerely,
Carla Lintas
Round 2
Reviewer 1 Report
Comments and Suggestions for Authors
The revised version of the Manuscript successfully addresses all criticisms raised during the initial peer review. No further comments.

Reviewer 2 Report
Comments and Suggestions for Authors
Dear Dr. Lintas and co-authors,
Thank you very much for your detailed and thoughtful responses to the reviewer comments, as well as for the substantial revisions made to your manuscript entitled “Identification of Candidate Genes for Endometriosis in a Three-Generation Family with Multiple Affected Members Using Whole Exome Sequencing.” I believe the revised manuscript now meets the standards for publication in Biomedicines. While the exploratory and preliminary nature of the findings is rightly emphasized, the study provides an important contribution to the field of familial endometriosis genetics.
Reviewer 3 Report
Comments and Suggestions for Authors
I appreciate the authors’ efforts in revising the manuscript titled "Identification of Candidate Genes for Endometriosis in a Three-Generation Family with Multiple Affected Members Using Whole Exome Sequencing." The revised version has improved in both clarity and structure. The addition of a workflow diagram and more detailed discussion have enhanced the manuscript’s readability and scientific rigor.
In particular, the authors have adequately addressed my previous concerns regarding the limited cohort size by elaborating on the limitations of small family-based studies and appropriately tempering their conclusions. The revised discussion is more cautious and better contextualized within existing literature.